# Individual Rubber Tree Segmentation Based on Ground-Based LiDAR Data and Faster R-CNN of Deep Learning

**Jiamin Wang [1,2,3], Xinxin Chen [1,2], Lin Cao [2], Feng An [4], Bangqian Chen [4], Lianfeng Xue [1] and Ting Yun [1,2,*]**

[1] College of Information Science and Technology, Nanjing Forestry University, Nanjing 210037, China; wjmin.wang@gmail.com (J.W.); cxx19950702@gmail.com (X.C.); xuelianfeng@njfu.edu.cn (L.X.)
[2] Co-Innovation Centre for Sustainable Forestry in Southern China, Nanjing Forestry University, Nanjing 210037, China; lincao@njfu.edu.cn
[3] School of Computer Software, Tianjin University, Tianjin 300350, China
[4] Danzhou Investigation and Experiment Station of Tropical Crops, Ministry of Agriculture, Rubber Research Institute, Chinese Academy of Tropical Agricultural Sciences, Danzhou 571737, China; bqchen@catas.cn (B.C.); an-f@catas.cn (F.A.)
*   Correspondence: njyunting@gmail.com; Tel.: +86-025-8542-7464

**Abstract:** Rubber trees in southern China are often impacted by natural disturbances that can result in a tilted tree body. Accurate crown segmentation for individual rubber trees from scanned point clouds is an essential prerequisite for accurate tree parameter retrieval. In this paper, three plots of different rubber tree clones, PR107, CATAS 7-20-59, and CATAS 8-7-9, were taken as the study subjects. Through data collection using ground-based mobile light detection and ranging (LiDAR), a voxelisation method based on the scanned tree trunk data was proposed, and deep images (i.e., images normally used for deep learning) were generated through frontal and lateral projection transform of point clouds in each voxel with a length of 8 m and a width of 3 m. These images provided the training and testing samples for the faster region-based convolutional neural network (Faster R-CNN) of deep learning. Consequently, the Faster R-CNN combined with the generated training samples comprising 802 deep images with pre-marked trunk locations was trained to automatically recognize the trunk locations in the testing samples, which comprised 359 deep images. Finally, the point clouds for the lower parts of each trunk were extracted through back-projection transform from the recognized trunk locations in the testing samples and used as the seed points for the region's growing algorithm to accomplish individual rubber tree crown segmentation. Compared with the visual inspection results, the recognition rate of our method reached 100% for the deep images of the testing samples when the images contained one or two trunks or the trunk information was slightly occluded by leaves. For the complicated cases, i.e., multiple trunks or overlapping trunks in one deep image or a trunk appearing in two adjacent deep images, the recognition accuracy of our method was greater than 90%. Our work represents a new method that combines a deep learning framework with point cloud processing for individual rubber tree crown segmentation based on ground-based mobile LiDAR scanned data.

**Keywords:** deep learning; tree crown segmentation; ground-based mobile LiDAR; rubber tree; Faster R-CNN

## 1. Introduction

Rubber (*Hevea brasiliensis* Muell. Arg.) trees, which are a widely planted hardwood genus in tropical areas, are important sources of natural rubber and wood. Hainan, as China's largest rubber production base, has nearly 8 million acres of rubber forest, forming the largest artificial ecosystem [1]. Due to its geographical location, Hainan Island's trees are frequently disturbed by typhoons and chilling injuries [2]. Typhoons that occur over a short period can cause serious damage, such as trunk and branch breakage and uprooting. Chilling damage is usually accompanied by long-term secondary damage of the rubber plantation, such as tree dieback, bark splitting, and bleeding [2]. To determine the wind resistance performance index of rubber trees and cultivate strong, resistant varieties, an accurate algorithm for individual rubber tree segmentation is indispensable for obtaining the structural parameters and dynamic change information of rubber trees of different clones [3].

The traditional acquisition of the structural parameters of rubber trees was executed via field measurements, but this process is very time consuming and labour-intensive and is useful only at the plot level as wells as being limited by small sample sizes and accessible areas. The rapid development of light detection and ranging (LiDAR) sensing methods provides a promising avenue for obtaining three-dimensional (3D) phenotype traits of plants with the ability to record accurate 3D laser points [4]. In terms of the carrying platform, laser scanning systems can be classified into five categories: satellite-based laser scanning (SLS), airborne laser scanning (ALS, namely, airborne LiDAR), mobile laser scanning (MLS), vehicle-borne laser scanning (VLS), and terrestrial laser scanning (TLS) [5]. Of these, SLS and ALS adopt a top-down scanning method, and MLS, VLS and TLS adopt a bottom-up scanning method. The method of top-down scanning can clearly scan the vegetation canopy, and it has great potential and advantages in recording the vertical structure characteristics of the forest and extracting the parameters of the canopy structure. The bottom-up scanning method can clearly record the lower part of the canopy (such as trunk and foliage), which is more suitable for ground forest survey work.

Based on the data obtained by the different scanning patterns mentioned above, a series of individual tree segmentation methods have been proposed and these methods can be divided into two categories according to the different scanning patterns: individual tree segmentation based on (1) ALS data and (2) TLS or MLS data. Based on ALS data, some efforts have been made to extract individual trees by extracting the characteristics of the tree organs in the upper tree parts (e.g., tree crown top or tree crown shape) through the calculation of the local maxima from either the canopy height model (CHM) [6] or scattered point clouds [7]. Image processing algorithms have been extensively used to segment individual tree crowns and locate local maxima to define the locations of tree tops, such as the mean-shift algorithm [8], K-means clustering [9], region growing algorithm [10], and watershed algorithm [11]. In addition, other concepts derived from computer science, such as voxelisation [12], the graph cut algorithm [13], adaptive size window filtering [14], the multilevel morphological active contour method [15], wavelet transform regarding time-frequency decomposition [16] and the topological relationship analysis method [17], have also been extended to delineate tree crowns from ALS data. For other methods based on TLS and MLS data, tree organs in the lower tree part (e.g., trunk) are taken as the basic elements to accomplish individual tree segmentation. For example, the random sample consensus (RANSAC) algorithm and Hough transform of circle detection have been adopted to recognize the horizontal cross section of a trunk and determine the trunk location [18,19]. The topological method (e.g., comparative shortest-path algorithm [20] or marked neighbourhood searching at voxel-scale from root points [21]) has been used to depict the structures of the non-photosynthetic components of trees, while least square fitting based on point clouds of trunks has been used to retrieve tree growth directions and the centres of tree crowns [22]. Finally, the revolving door schematic mode of the morphological method [23] has been used to automatically recognize tree crowns from MLS data while simultaneously precluding the interference from poles and buildings.

Although scholars have proposed many methods for individual tree segmentation, accurate individual tree crown segmentation based on ALS and MLS data is still needed for further improvement, especially for ecological forests in which tree crowns can be extremely irregular and

are often heavily intersected. Meanwhile, two separate issues have arisen concerning (1) the deformation of the wood structure of the studied trees induced by exposure to perennial hurricane disasters and (2) the general circles detected by existing methods (i.e., Hough transform) are not suitable for some trunks with elliptical or irregular shapes of horizontal cross sections or complications stemming from wild-grown twigs or leaflets originating from the trunk. Hence, the robustness, generality and ability of existing models to provide accurate individual tree segmentation from ground-based mobile LiDAR data need further study.

Deep learning is a new area of machine learning that originates from artificial neural networks. Different from machine learning methods, such as support vector machines [24] or neural networks [25], deep learning has a more complex network structure and utilises multiple sizes of convolution temples to perform the task of intrinsic image decomposition and accomplish the iterative backward or forward propagation for adjusting the weights of neurons. Meanwhile, deep learning designs prediction bounding boxes (anchors) and compares these boxes with ground truth data derived from numerous training samples, and various gradients of the loss function are evaluated to automatically recognize target locations. With a special network model structure, deep learning methods learn the internal features of massive images, which can enhance the image recognition and classification accuracy and stability, compared with the extraction of single features in artificially designed programs [26]. The convolutional neural network (CNN) [27], as the most commonly method applied to analyse visual imagery in the deep learning framework, uses relatively little pre-processing compared to other image classification algorithms and learns the internal features of massive images rather than single artificially designed features. Thus, CNNs achieve state-of-the-art performance in some image-based phenotypic recognition tasks, such as classifying vegetables [28], recognizing the maturity levels of tomatoes [29], segmenting individual maize plants [30], and detecting plant disease [31].

In this paper, a novel individual tree segmentation method combined with the deep learning method of Faster R-CNN is proposed. The research objectives of this paper mainly include (1) data collection for three rubber tree forest plots of different clones using ground-based mobile LiDAR; (2) application of a voxelisation method and projection strategy to transform the 3D scanned points of the trunks of the rubber trees to deep images (namely, the images generated by the projection strategy for Faster R-CNN of deep learning); (3) location of rubber tree trunks on many deep images generated from voxelisation of scanned points of many rubber trees by training and testing the Faster R-CNN; and (4) segmentation of individual trees based on detected scanned trunk points and the region growing algorithm. Our approach uses the concept of deep learning, which provides a strong identification ability through large training sample analysis to automatically recognize the trunk information without being affected by a tilted tree trunk or wild-grown twigs or leaves. The workflow of our method is shown in Figure 1.

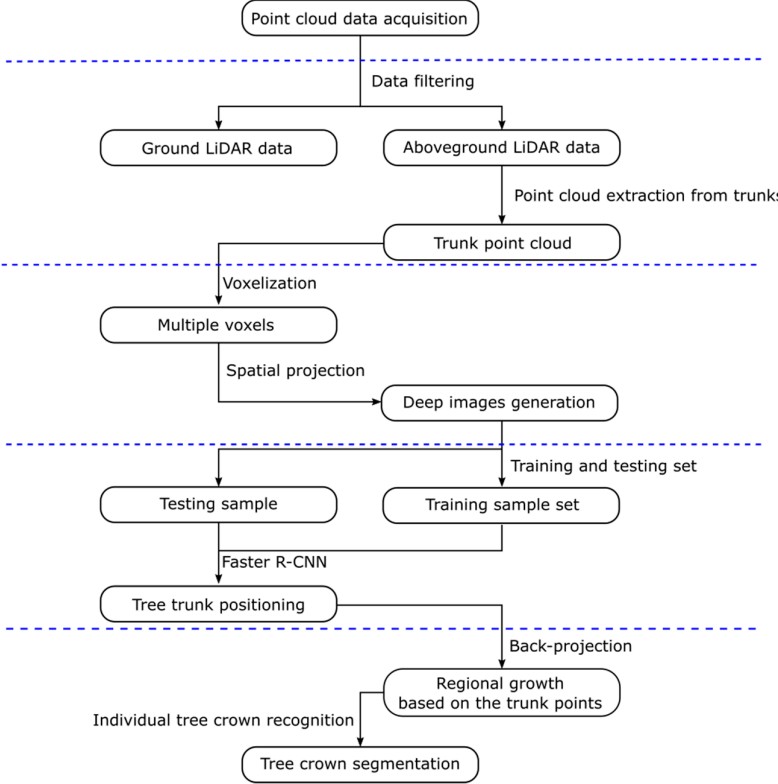

**Figure 1.** An overview of the workflow for the individual tree segmentation method based on deep learning.

## 2. Materials and Methods

### 2.1. Study Area

The study area is located in the experimental station of the Rubber Research Institute of the Chinese Academy of the Tropical Agricultural Sciences (19°32′47.89″ N, 109°28′29.33″ E) in Danzhou city, Hainan Province (Figure 2). As China's largest rubber production base, Hainan Island has a tropical monsoon climate with a rainy season that lasts from May to October and a dry season that lasts from November to April. The average annual precipitation is 1815 millimetres, and the annual mean temperature is approximately 23.2 °C. This climate is favourable for agricultural development, and the cultivation of rubber trees is continuously increasing in this area. The plantation has reclaimed over 5000 ha of cultivated land and tropical rainforest since it was established in 1957. Over the past sixty years, more than 100 hurricanes have hit Hainan Island, and though a hurricane occurs over a short period, these storms cause serious primary damage, such as trunk and branch breakage and uprooting. As shown in Figure 2, three tree clones, including rubber tree PR107 (rubber forest plot 1), rubber tree CATAS 7-20-59 (rubber forest plot 2) and rubber tree CATAS 8-7-9 (rubber forest plot 3), in the rubber tree plantation were chosen as typical trees for our experiments.

Three subsets from the three rubber forest plots were created and used as the training sites in the follow-up experiment. Each subset consisted of an approximately 0.6 × 0.6 km area that was representative of the corresponding rubber forest plot, and the tree height measurements within the three subsets were conducted on 11 February 2016 using a Vertex IV hypsometer (Haglöf, Långsele, Sweden). In addition, we selected another three subsets from the three forest plots (which did not intersect with the subset used as the training sites), and each subset was composed of an area of approximately 0.3 × 0.6 km, which were used as the testing sites.

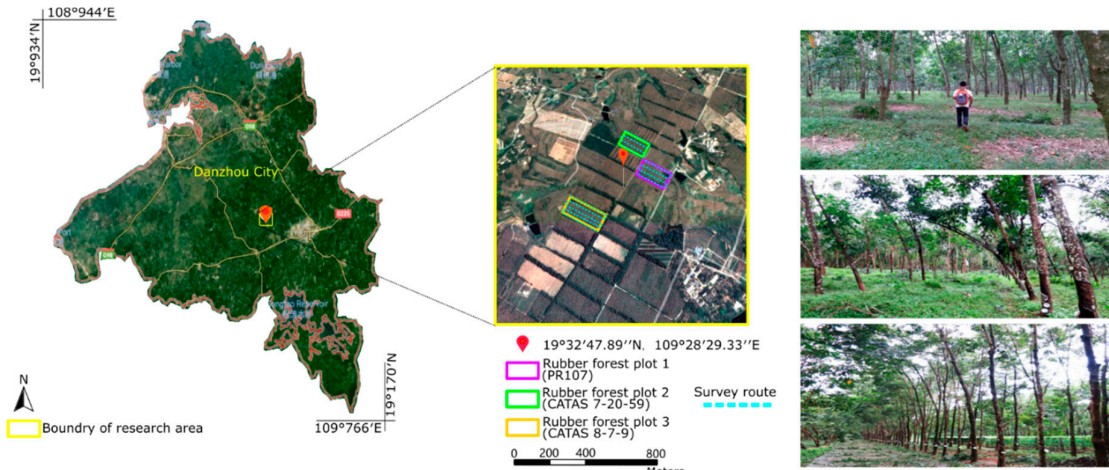

**Figure 2.** General situations of the study area. Left: The location of the study area and the three forest plots of rubber trees within the experimental farm, Danzhou, Hainan Island, China. Middle: The remote sensing image acquired from Google Earth, where the different coloured rectangles mark the edges of the different rubber forest plots, and blue dashed lines represent the survey routes using ground-based light detection and ranging (LiDAR). Right: The photos show our scanning process in the rubber forest plots using man-portable mobile LiDAR.

### 2.2. Laser Data Acquisition

The LiDAR data were measured on 10 October 2016, using a Velodyne HDL-32E laser scanner (Table 1). The Velodyne HDL-32E scanning system was deployed by an experimenter, and the scanner was set to "continuous shooting mode" to collect data. The survey route was programmed to follow a predetermined rectangular parallel plan designed to cover the three study sites, and the experimenter traced the survey lines (dashed blue lines in Figure 1) at a speed of approximately 0.5 m/s due to the complex terrain of the rubber tree plantation and because the experimenter was required to carry a heavy scanning instrument. As the Velodyne HDL-32E sensor continued to emit laser beams in work mode, while the experimenter traced the survey lines, more scanning perspective views are obtained and the density of the scanned data are higher when the platform runs slower while loading the laser scanner. The Velodyne uses LiDAR technology, and together with the real-time referencing of laser returns provided by the GPS, this system automatically generates point clouds with spatial coordinates. The Velodyne LiDAR system integrates laser scanning with SLAM (simultaneous localization & mapping) technologies [32] to rapidly complete the registration of each scan and generate a high-density point cloud for each target rubber forest plot.

**Table 1.** Specification of Velodyne HDL-32E.

| Device | Technical Parameters | Technical Specification |
|---|---|---|
| Velodyne HDL-32E | Scanner weight | <2 kg |
| | Operating temperature | −10 °C to +60 °C |
| | Field of view | Horizontal: 0° to 360° Vertical: −30.67° to +10.67° |
| | Scanning accuracy | <2 cm |
| | Points per second | Up to 700,000 |
| | Laser wavelength | 905 nm |
| | Scanning frequency | 10 Hz |

### 2.3. Data Pre-Processing

The point cloud data from the experimental plots scanned by the laser scanner were first classified as aboveground points and ground points using a cloth simulation filtering (CSF) [33]

ground filtering method. The rubber tree canopy often intersected, which seriously affected the region's growth results. Therefore, in this study, the wood-leaf separation operation [34] was performed on the aboveground points of experimental plots, which aimed to classify LiDAR points into wood and leaf components. Figure 3 shows the magnified wood-leaf separation results of several typical trees belonging to the three rubber tree clone types, where the classified leaf points are shown in green and the classified wood points are shown in crimson.

The artificial rubber forests studied in this paper have similar tree age, plant spacing and stand structure. Therefore, it is relatively easy to calculate the crown base height. To generate the two-dimensional images of the trunks, a voxelisation operation is carried out on the point cloud of the trunks of these three training sites. The planting spacing of artificial rubber forests was generally approximately 6–8 m and 2.8–3 m in rows and lines, respectively. Therefore, during voxelisation, we define voxels with a length of 8 m and a width of 3 m. Meanwhile, different heights of voxels were set for the three training sites according to the crown base height of the three rubber tree clones.

After voxelization, all the laser points of the three training sites were assigned to the corresponding voxels $V$ . As shown in Figure 3, for each voxel, $vi(vi \in V)$ , two corresponding deep images were generated by projecting from the Y-axis positive direction and the X-axis positive direction. The number of generated deep images for the scanned points from the three training sites were 233, 268 and 301 (Table 2). All of these images were used to construct the training set for training the Faster R-CNN [35] model to learn the rubber tree trunk locations in deep images.

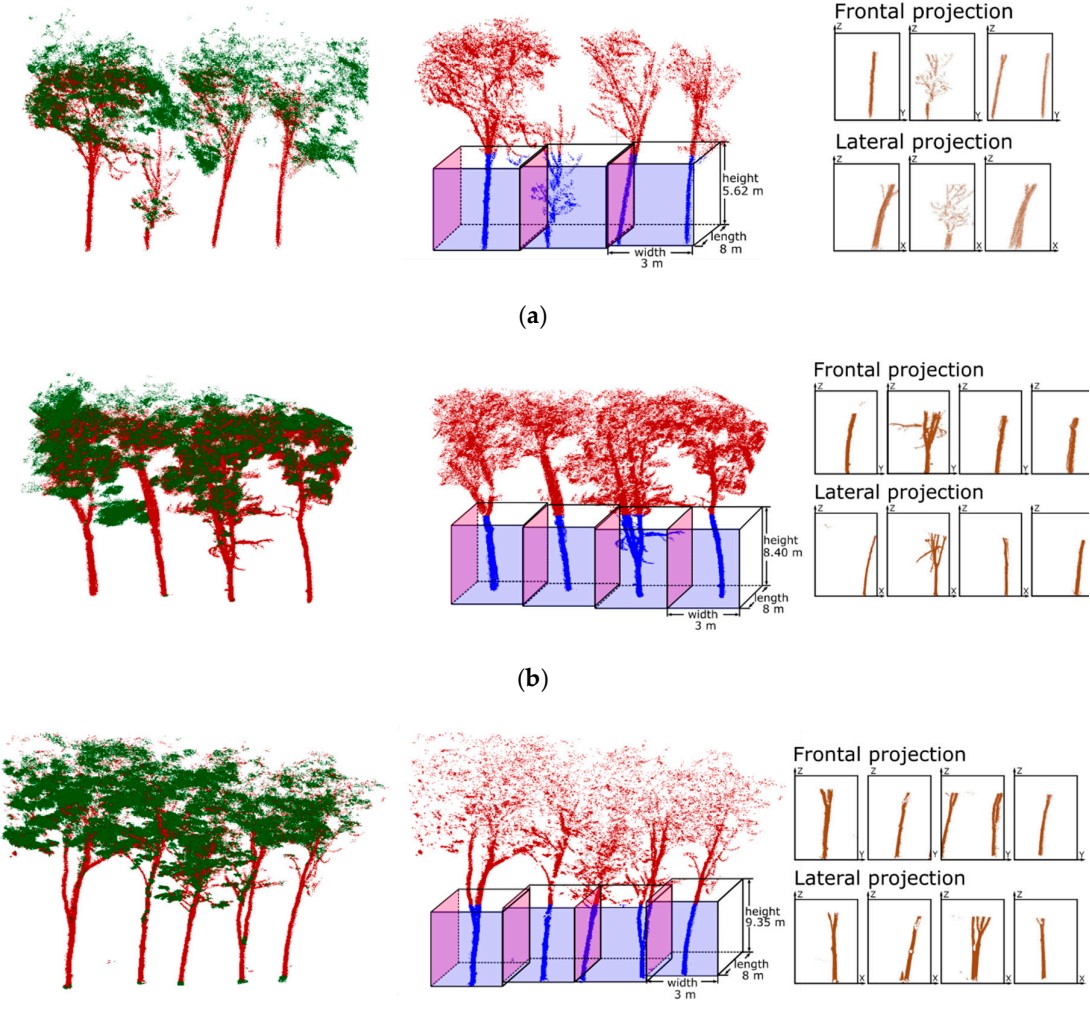

**Figure 3.** Data pre-processing and multi-angle projecting results for the three rubber forest plots: (**a**) clone PR107, (**b**) CATAS 7-20-59 and (**c**) CATAS 8-7-9. Left: Wood-leaf classification results of scanned data of the typical trees belonging to the three forest plots, where the classified leaf points shown in green and the classified wood points shown in crimson. Middle: Voxelisation of the typical trees from three forest plots, where the length and width of the voxel are set according to the actual plant spacing and the height of the voxel is set according to the crown base height of the three rubber tree clones. Right: Deep images generated using frontal and lateral projection from the scanned points of lower part of the trees in three forest plots.

**Table 2.** Detailed description of the collected data and algorithm-related parameters.

| | | **Rubber Tree Plot 1 (PR107)** | **Rubber Tree Plot 2 (CATAS7-20-59)** | **Rubber Tree Plot 3 (CATAS8-7-9)** |
|---|---|---|---|---|
| Number of scanned points/number of trees | | 5,387,676/180 | 7,097,159/256 | 8,820,133/276 |
| Average tree height (m) | | 15.97 | 17.11 | 16.05 |
| Training sites | Number of scanned points/trees number | 3,711,510/124 | 4,879,297/176 | 6,039,974/189 |
| | Length/width/height of voxels (m) | 8/3/5.62 | 8/3/8.40 | 8/3/9.35 |
| | Number of generated deep images | 233 | 268 | 301 |
| Testing sites | Number of scanned points/trees number | 1,676,166/56 | 2,217,862/80 | 2,780,159/87 |
| | Number of generated deep images | 90 | 126 | 143 |

### 2.4. Faster R-CNN

Faster R-CNN is adopted to recognise the location of tree trunks from every deep image and is composed of two CNNs, i.e., a region proposal network (RPN) that proposes regions and a fast region-based CNN (Fast R-CNN) detection network that uses the proposed regions. The RPN samples the information from a random region in the image as the proposal regions and trains them to determine the areas that may contain the target [36]. The Fast R-CNN detection network further processes the area information collected by the RPN network, determines the target category in the area, and precisely adjusts the size of this area to locate the specific location of the target in the image [26]. Figure 4 illustrates the architecture of the automatic rubber tree trunk detection and recognition model, which is based on the Faster R-CNN model.

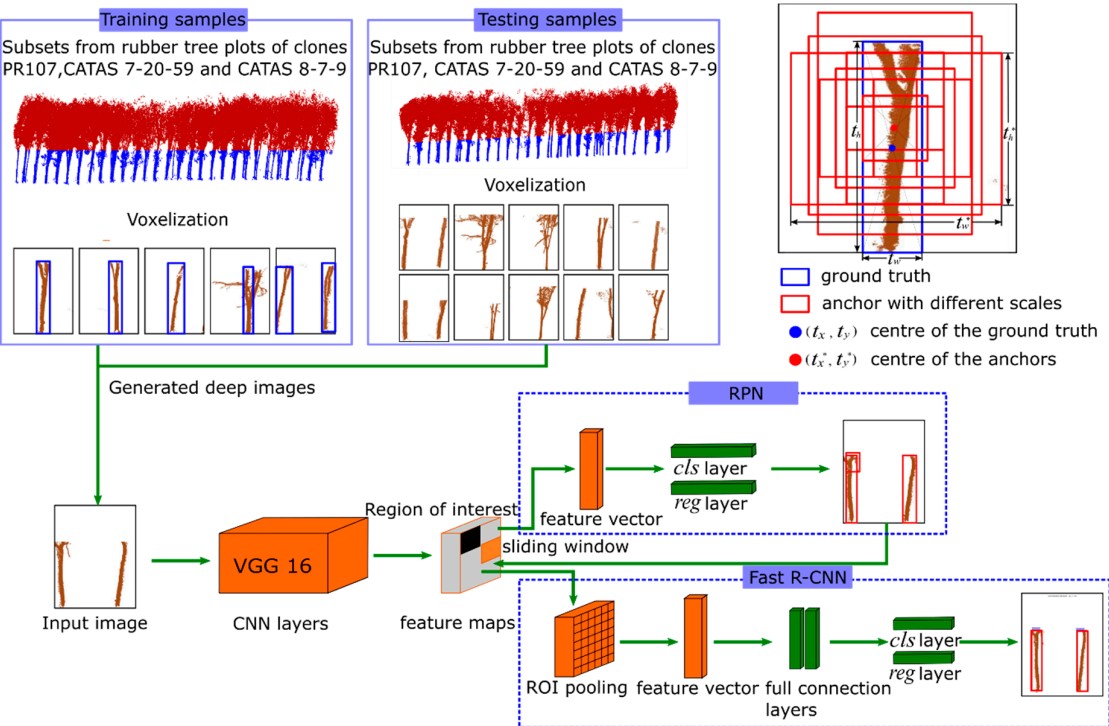

**Figure 4.** The flow chart of the Faster R-CNN (faster region-based convolutional neural network) used to detect the rubber tree trunk in deep images. The deep images for the training samples generated from the subsets of three rubber tree plots were fed into the Faster R-CNN to classify and regress the anchor (bounding box) of the trunk in deep images. The deep images of the testing samples generated from other subsets of three rubber tree plots were tested to obtain the anchor of the trunk by the trained Faster R-CNN.

The training process mainly consists of four steps. First, the parameters of the whole Faster R-CNN network were initialised with the pre-trained model, and then the RPN network was trained with our training set. Second, the proposed region generated by the trained RPN was used to train the Fast R-CNN detection network. Finally, the RPN and the Fast R-CNN constituted a joint network, and the weight of the joint network was tuned by repeating the above process until the training loss reached a threshold [26].

Based on the pre-trained CNN, the deep images including the rubber tree trunks were constructed as a training set to optimise the parameters of the Faster R-CNN. As described in section 2.3, the training samples were projected from the voxels, and according to the number and state of the rubber tree trunks in the images, the images were divided into 6 cases for analysis: (a) the images that contained only one complete tree trunk; (b) the images that contained two complete tree trunks; (c) the images that contained multiple trunks with branches that appeared in a voxel; (d) the images in which the information of the trunk was occluded by leaves or branches; (e) the images in which the information of trunks belonging to multiple trees overlapped in one voxel; (f) the images in which the trunk of a tree appears in two adjacent images generated by the projection of two adjacent voxels. To prepare for the follow-up training processes, the training sample must be considered. Note that the marking methods for the six cases were discussed separately: (a) mark the whole tree trunk; (b) mark the trunks of the two trees in one image; (c) mark the trunks of the two trees in one image, including the branches of the trees; (d) mark only the tree trunks, do not mark the leaves; (e) mark all the overlapping trees; (f) mark only trunks in the voxel and do not mark the upper part of the trunk and branches that appear in the adjacent voxel. Figure 5 shows the typical results of the training images, where the blue rectangular bounding boxes marked by us that tightly surround the target were used as the ground truth in the follow-up training.

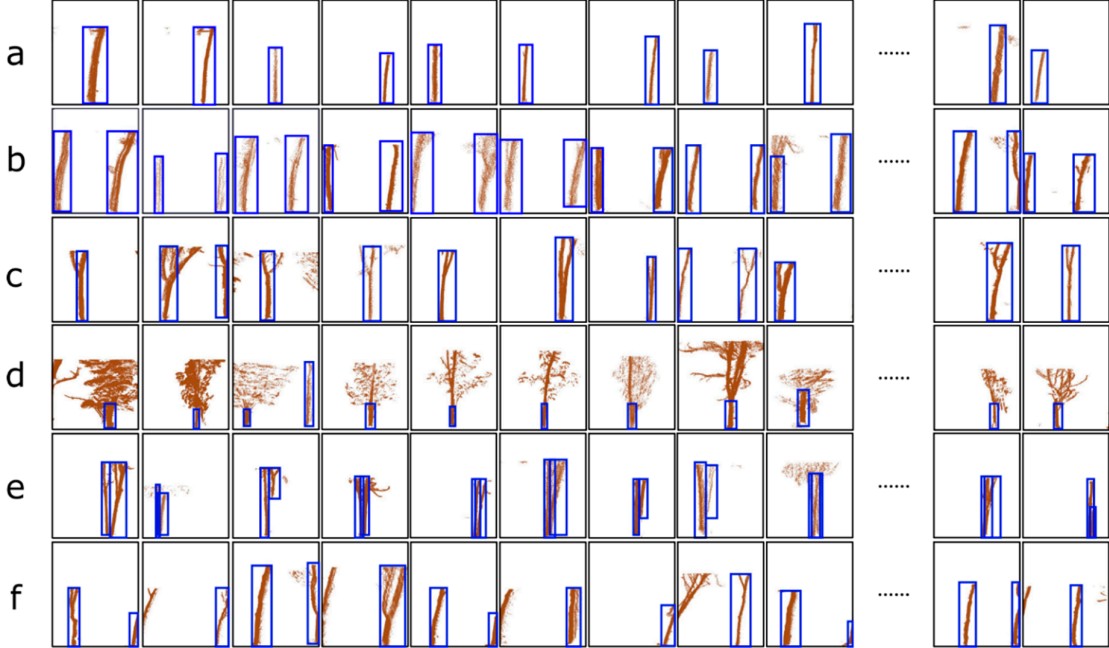

**Figure 5.** Six examples of the training samples comprising generated deep images: (**a**) the images contain only one complete tree trunk; (**b**) the images contain two complete tree trunks; (**c**) the images that contain multiple trunks with branches that appear in a voxel; (**d**) the images in which the information of the trunk is occluded by leaves or branches; (**e**) the images in which the information of the trunks belonging to multiple trees overlap in one voxel; (**f**) the images in which the trunk of a tree appears in two adjacent images generated by the projection of two adjacent voxels. Different trunk identification methods for these six complicated cases form the training sample: (**a**) mark the whole tree trunk; (**b**) mark the trunks of two trees in one image; (**c**) mark the trunks of two trees in one image, including the branches of the trees; (**d**) mark only the tree, do not mark the leaves; (**e**) mark all the overlapping trees; (**f**) mark only trunks in the voxel, do not mark the upper part of the trunk and branches that appear in the adjacent voxel. All the blue rectangular bounding boxes tightly surround the target trunks, which is called the ground truth in the follow-up training.

### 2.4.1. The Pre-Training CNN Model

As shown in Table 2, the total number of deep images generated to construct the training set was 802, but a large number of samples is required for high-precision training [26]. Therefore, the transfer learning method was used to solve this problem [37]. The transfer learning method can use common data to obtain a pre-trained model to construct an RPN network and detection network. In this paper, the training set (approximately 100,000 images, 1000 classes) in ImageNet (first published at the Computer Vision and Pattern Recognition (CVPR) conference (2009)) [38] was used to pre-train the VGG16 network model [39]. The VGG16 network was the feature extraction network used to extract the image features.

### 2.4.2. Training Process of the RPN Network

As an important part of the Faster R-CNN, the RPN neural network takes an image as input and generates a set of rectangular proposal regions. Its special network structure can promote the region extraction speed.

With the convolution layer of the feature extraction network, the feature map of the input image was generated. Then, for each position of the feature map, a convolution operation was performed via a 3 × 3 sliding window to obtain the multidimensional feature vector corresponding to the same position, which reflected the deep features in the small positioning window. This feature vector was fed into two related and fully connected layers: a classification (cls) layer and a regression (reg) layer. Anchors [35] (i.e., multiple bounding boxes with different sizes and aspect ratios centred on each

pixel used to predict the ideal location, shape and size of target trunks through calculation of the degree of overlap with the ground truth data) were centred in the 3 × 3 sliding window. In this paper, each sliding window had k = 9 anchors with a combination of three scales (128², 256², 512²) and three aspect ratios (1:1, 1:2, 2:1) (see Figure 4). If an anchor had an overlap area higher than 0.7 with any ground truth, we assigned a positive label to this anchor (i.e., positive anchor). In some rare cases, the above condition found no positive anchor; at this time, we specified the anchor that had the highest overlap area with a ground truth as a positive anchor. We assigned a negative label to a non-positive anchor if its overlap area was lower than 0.3 with all ground truths. Then, the probability value was used to determine whether each anchor belonged to the foreground (i.e., the information in the anchor was recognised as the target) or the background (i.e., the information in the anchor was recognised as the non-target), and the position deviation of the anchor relative to the ground truth was correspondingly generated. When the information in the anchor was recognised as the target, the anchor was reserved as the proposal region and used for subsequent training. Before training the Faster R-CNN detection network, the proposed regions generated by the RPN network were mapped to the feature map, and a series of regions of interest (ROIs) with random size were generated on the feature map (see the black patch in Figure 4).

### 2.4.3. The Training of the Fast R-CNN Detection Network

The ROI with random size was chosen as input for the ROI pooling layer (see Figure 4), and through the ROI pooling, the ROI was normalised into a fixed size. The ROI with fixed size was used to calculate each proposal specific category through the two full connection layers. One full connection layer was used to realise classification for the target in the ROI. The other full connection layer was used to justify the position of the corresponding proposal region by the regional regression. After adaptive correction by Fast R-CNN, the position of the proposal regions was adjusted to the ground truth.

### 2.4.4. The Loss Function of the Training Process

In the training process, the parameters of the neural network were adjusted by the loss function [35], which is defined as follows:

$$L(\{e_i\},\{t_i\}) = \frac{1}{N_{cls}}\sum_i L_{cls}(e_i,e_i^*) + \lambda\frac{1}{N_{reg}}\sum_i e_i^* L_{reg}(t_i,t_i^*),\tag{1}$$

where $i$ is the index of an anchor. The classification loss $L_{cls}(e_i,e_i^*)$ is the log loss function for two classes (foreground or background).

$$L_{cls}(e_i,e_i^*) = -\log[e_i^* e_i + (1-e_i^*)(1-e_i)],\tag{2}$$

where $e_i$ is the probability that anchor $i$ belongs to the foreground and $e_i^*$ is the ground truth label. If the anchor $i$ is positive (i.e., an overlap area in the anchor with ground truth higher than 0.7), the ground truth label $e_i^*$ is assigned to 1. If the anchor $i$ is negative (i.e., an overlap area in the non-positive anchor with ground truth lower than 0.3), the $e_i^*$ is assigned to 0.

The regression loss function is

$$L_{reg}(t_i,t_i^*) = R(t_i - t_i^*),\tag{3}$$

For anchor $i$, if the $e_i = 0$, the regression loss function is disabled; if the $e_i^* = 1$, the regression loss function is activated. $t_i = \{t_x,t_y,t_w,t_h\}$ is a vector that represents the coordinates of the anchor whose size and position are constantly changing during iterative training. $t_i^* = \{t_x^*,t_y^*,t_w^*,t_h^*\}$

represents the coordinates of the ground truth associated with anchor $i$ (see Figure 4). In this experiment, smooth $L1$ is used here to substitute the function $R$ for loss function $L_{reg}$.

$$L_{reg}\left(t_i, t_i^*\right) = R\left(t_i - t_i^*\right) = smooth_{L1}\left(x\right) = \begin{cases} 0.5\left(t_i - t_i^*\right)^2 & if \left|t_i - t_i^*\right| < 1 \\ \left|t_i - t_i^*\right| - 0.5 & otherwise \end{cases}, \tag{4}$$

The outputs of the *cls* and *reg* layers (see Figure 4) are composed of $\{e_i\}$ and $\{t_i\}$, respectively. $N_{cls}$ is the size of the feature map (approximately 1750, 50 × 35), and $N_{reg}$ is the batch size (in the RPN network, $N_{reg} = 256$; in the Fast R-CNN detection network, $N_{reg} = 128$). The parameter $\lambda$ is used to balance the $L_{cls}$ and $L_{reg}$, so that the total loss function $L$ can be considered as two kinds of loss.

Using the marked rubber tree trunks as references, the difference between the predicted region information and the ground truth was generated. Then, the backpropagation algorithm was used to tune the weight and offset of the network. With enough training, the Faster R-CNN can detect the accurate position and identification of the rubber tree trunks from the deep images.

2.4.5. The Testing Process for Using Faster R-CNN to Recognise Trunks

The testing process included four main steps. For each testing site, the point clouds were assigned to different voxels by voxelization. Second, the voxels were used to generate deep images through frontal and lateral projection transform. Third, the 359 generated deep images belonging to the testing samples were analysed by the trained network model to predict the locations of the trunks, and only the predicted results with more than 90% prediction confidence were retained. Finally, the location of the voxel and the four coordinates of the predicted bounding box of the trunks in each image were recorded and used to perform a back-projection transform to extract the corresponding scanned trunk points. These extracted trunk points for each rubber tree were taken as the seed points for region growing [40] to extract individual tree skeletons. Based on the extracted tree skeleton of each rubber tree, the unsegmented leaf points were classified into the corresponding rubber tree skeletons by using the clustering algorithm.

The individual tree crown segmentation results were evaluated versus the visual inspection results for all three rubber forest plots. *TP* (true positive) was the number of correctly detected trees, *FN* (false negative) was the number of trees that were not detected (omission error), and the *FP* (false positive) was the number of extra trees that did not exist in the field (commission error). We expected high *TP*, low *FN*, and low *FP* values to result in high accuracy. Moreover, the tree detection rate r (recall), the correctness of the detected trees *P* (precision), and the overall accuracy of the detected trees *F* (*F*-score) for each site were calculated using the following equations [41].

$$r = \frac{TP}{TP + FN}, \tag{5}$$

$$P = \frac{TP}{TP + FP}, \tag{6}$$

$$F = 2 * \frac{r * P}{r + P}, \tag{7}$$

## 3. Results

### 3.1. Testing the Faster R-CNN Model

The "end-to-end" CNN was built based on the TensorFlow [42] of deep learning framework, and the experiments were performed on a PC with an Intel i7-8550U CPU, 16 GB RAM and a NIVIDA GTX 1070 GPU.

The Faster R-CNN training is carried out using the momentum method. We use a weight decay of 0.0001 and a momentum of 0.9. The learning rate is 0.001, and the number of iterations is 70,000. The training loss is plotted in Figure 6, and the total training time is approximately 2 h.

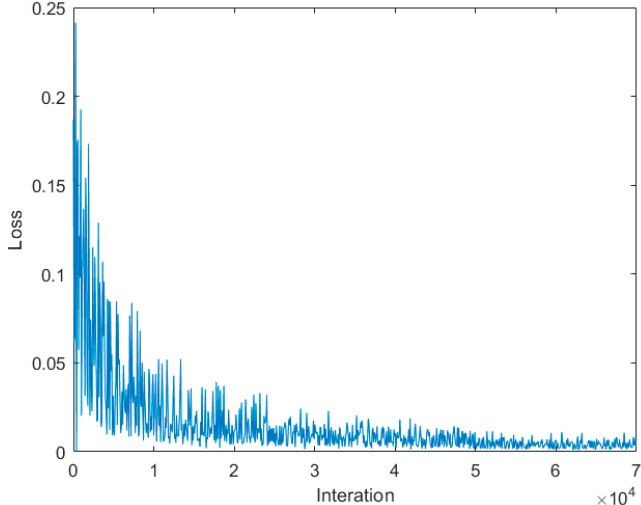

**Figure 6.** The curve of training loss value of Faster R-CNN for trunk detection from the deep images. Local fluctuations of the curve were caused by repeatedly confirming the sizes of the bounding boxes to accurately delineate the tree trunks from the deep images, but the overall downward trend of the curve indicates a better convergence result of training.

Although the training loss was not sufficiently smooth, the overall declining trend was obvious (see Figure 6). The loss declined mainly in the first 100 iterations. The final loss was approximately 0.002, which represents the small error between the predicted results and the corresponding ground truths. Meanwhile, the strong loss fluctuation during training was due to the instability in the process of accurately seeking the locations of anchors (i.e., bounding boxes seen in Figure 4) based on gradient descent. During the training process, the network encountered some inaccurate bounding boxes from which it could not learn effective features of rubber tree trunks, resulting in the strong fluctuations in the value of the regression loss function. Despite this phenomenon, the loss function value and the intensity of the fluctuation in the regression loss function gradually decreased as the number of iteration increased, which resulted in better convergence of the training results.

Figure 7 shows the typical correctly labelled, missing labelled and incorrectly labelled results from Faster R-CNN for the rubber tree trunks on the deep images belonging to the testing samples for the six cases mentioned in section 2.4.

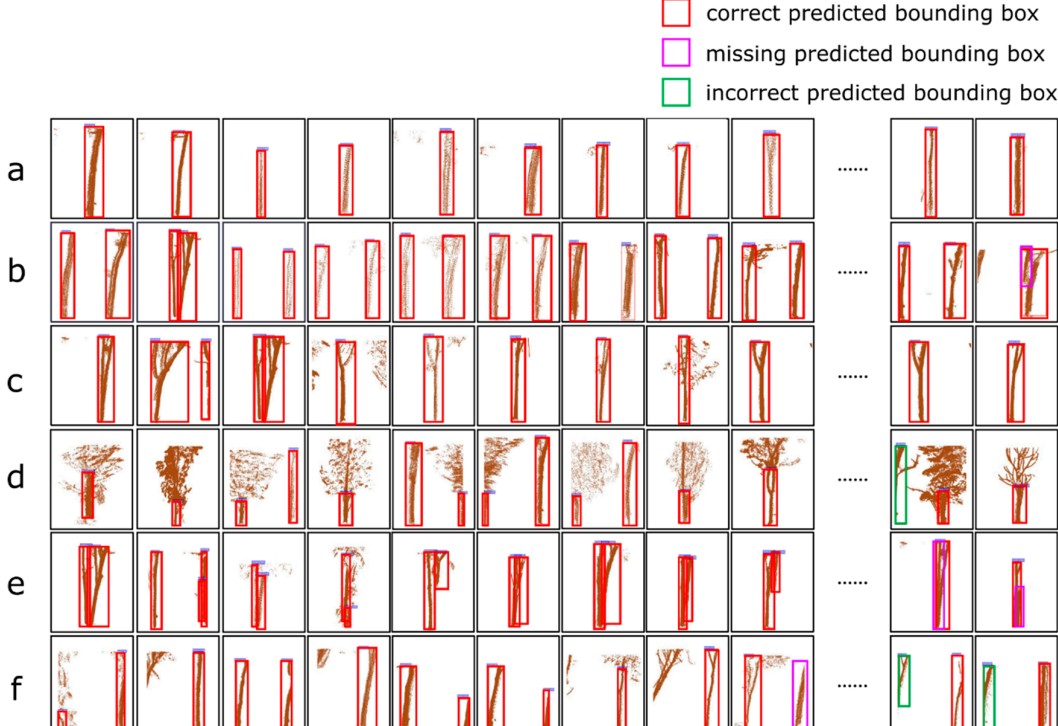

**Figure 7.** Recognition of tree trunks in deep images belonging to the testing samples and analysis of the results corresponding to the six cases: (**a**) shows the accurate labelling of the tree trunks using our algorithm for the images containing only one complete tree trunk; (**b**) shows that our algorithm labels the multi-trunk parts in one image, instead of the leaves; (**c**) shows that our algorithm labels the multi-trunk part and the intersection of the branches in one image; (**d**) shows the accurate labelling of the tree trunks using our algorithm, instead of the upper branch of the trunks and leaves; (**e**) shows the labelling of tree trunks using our algorithm for images containing overlapping tree trunks, where the trunks closer to us are successfully labelled, and those farther away are not labelled by the algorithm; and (**f**) shows the labelling of tree trunks using our algorithm for a single tree trunk appearing in two adjacent deep images, where a small amount of the lower part of the trunk is missing labelled and a small amount of the upper part of the trunk is incorrectly labelled.

### 3.2. Realising Individual Tree Segmentation

After recognising the locations of the tree trunks in each testing image, a back-projection transform from the trunk locations in the deep images to spatial scanned trunk points was adopted, and the extracted trunk points of each rubber tree were taken as the seed points to extract individual tree skeletons using the region growing method [40]. The specific operation based on the region growing principle was to first search the skeleton points that were related to the seed points and then calculate the distances between the skeleton points and the seed points. If the shortest point-to-point distance was smaller than a threshold and the skeleton points satisfied the continuity condition with the seed points, the skeleton point was added to the linked list of the seed points. The iterative process repeated until no new points were added. Based on the extracted tree skeleton points of each rubber tree, the un-segmented leaf points were classified into the corresponding rubber tree skeleton by using the nearest neighbour clustering algorithm [43] to complete individual tree segmentation. The individual rubber tree trunk positioning results for the three rubber forest plots are shown in Figure 8. The individual rubber tree trunks are represented by different colours, which serve as seed points to further segment the point cloud (represented in crimson). The region growing results are shown in Figure 9, and the results of individual tree crown segmentation using the nearest neighbour clustering algorithm are shown in Figure 10.

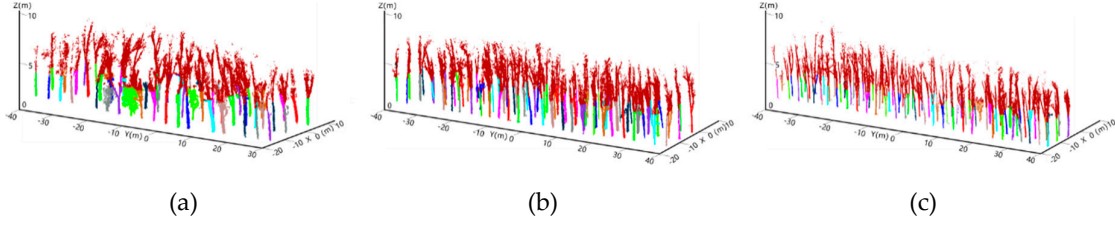

(a)                    (b)                    (c)

**Figure 8.** Program diagrams showing the results of the scanned trunk point detection using back projection transform from the detected tree trunk location in each deep image using Faster R-CNN, where the detected lower parts of each trunk are indicated by different colours. (**a**), (**b**) and (**c**) show the detected results for the subset from rubber tree plot 1 (PR 107), rubber tree plot 2 (CATAS 7-20-59) and rubber tree plot 3 (CATAS 8-7-9).

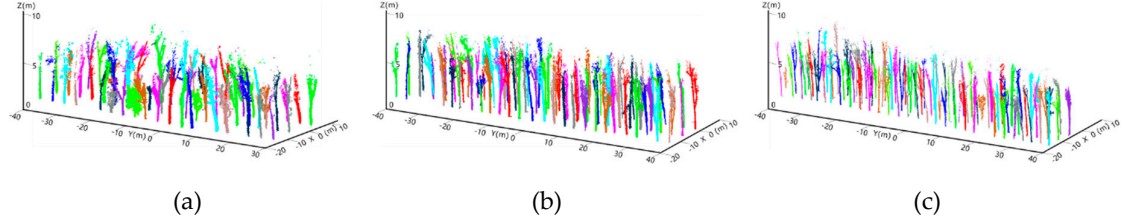

(a)                    (b)                    (c)

**Figure 9.** Taking the scanned points of the detected lower parts of each trunk as the seed points, program diagrams showing the results of the regional growing, the extracted skeletons of each tree related to the corresponding seed points are indicated by different colours. (**a**), (**b**) and (**c**) show the regional growing results for the subset from rubber tree plot 1 (PR 107), rubber tree plot 2 (CATAS 7-20-59) and rubber tree plot 3 (CATAS 8-7-9).

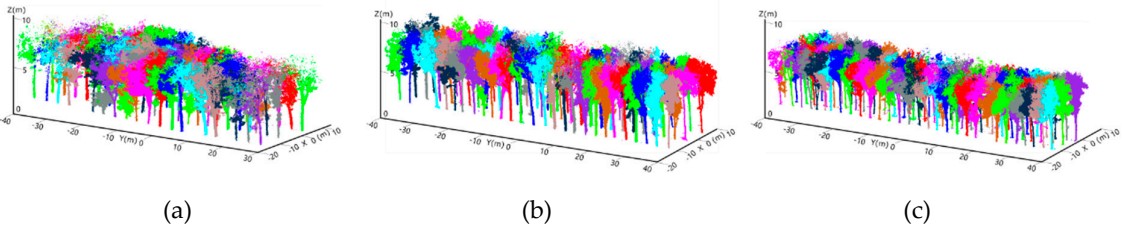

(a)                    (b)                    (c)

**Figure 10.** Program diagrams showing the results of the leaf point cloud clustering based on every extracted tree skeleton to complete individual tree crown segmentation, where different colours indicate the segmentation results for each rubber tree. (**a**), (**b**) and (**c**) show the segmentation results for the subset from rubber tree plot 1 (PR 107), rubber tree plot 2 (CATAS 7-20-59) and rubber tree plot 3 (CATAS 8-7-9).

Sound performance was achieved in the tree segmentation results for the three rubber forest plots using our method (Figure 10). The overall values of *r*, *P*, and *F* were all 0.99. For forest plot 1 (PR 107), the values of *r*, *P*, and *F* were 1, 0.98, and 0.99, respectively. For forest plot 2 (CATAS 7-20-59), the values of *r*, *P*, and *F* were 0.99, 0.99, and 0.99, respectively. For forest plot 3 (CATAS 8-7-9), the values of *r*, *P*, and *F* were 0.98, 0.99, and 0.98, respectively (Table 3). These segmented accuracies were almost the same, indicating that our method based on deep learning was not susceptible to variation in the rubber tree clones.

**Table 3.** Accuracy assessments of the individual rubber tree segmentation on the three rubber forest plots with different rubber tree clones.

| | Number of Trees/Images | Number of Segmented Trees | *TP* | *FP* | *FN* | *r* [1] | *P* [2] | *F* [3] |
|---|---|---|---|---|---|---|---|---|

| Forest plot 1 (PR 107) | 56/90 | 56 | 56 | 1 | 0 | 1 | 0.98 | 0.99 |
|---|---|---|---|---|---|---|---|---|
| Forest plot 2 (CATAS 7-20-59) | 80/126 | 78 | 78 | 1 | 2 | 0.98 | 0.99 | 0.98 |
| Forest plot 3 (CATAS 8-7-9) | 87/143 | 85 | 85 | 1 | 2 | 0.98 | 0.99 | 0.98 |
| Overall | 223/359 | 219 | 219 | 3 | 4 | 0.98 | 0.99 | 0.98 |

[1] *r* (recall): tree crown detection rate. [2] *P* (precision): the correctness of the detected tree. [3] *F* (*F*-score): the overall accuracy of detected tree.

To detail the location errors in the segmentation results, the testing set was divided into six cases mentioned in Figure 5 according to the number and state of the rubber tree trunks contained in each deep image, and the number of the deep images in the six cases were 228, 22, 48, 18, 14, and 29, and the accuracy of rubber tree trunk detection in these six cases was calculated. For each case mentioned in Figure 5, we computed the mean recall *r*, mean precision *P*, mean *F*-score for the comparison of the results. As shown in Table 4, for cases a, c, and d, the values of *F* reached 100%. For cases b, d, and e, although the *FP* and *FN* obtain small value, the value of *F* is higher than 90%.

Certain factors may lead to the detection errors in the above three cases (b, d and e). First, as shown in Table 4, there are relatively few deep images used to train the three cases (b, d and e), which may lead to under-fitting problems (i.e., deficiently irrelevant deep images were used for training, resulting in low testing result accuracy). In addition, for case e, multiple trunks may have a high degree of overlap after projection, which may cause serious interference with trunk detection. For case f, the crooked rubber tree trunks caused by long-term hurricane disturbances may have been voxelised into two voxels. When marking the corresponding training samples, only the lower part of these trunks was marked in the deep images. However, due to the similar shape characteristic of trunks, the upper parts of the trunks detected by the trained network were considered incorrect.

**Table 4.** Accuracy assessment of individual rubber tree segmentation of rubber trees in different cases.

| | Training Sites | Testing Sites | | | | | | | |
|---|---|---|---|---|---|---|---|---|---|
| | Number of Trees/Images | Number of Trees/Images | Number of Detected Trees | *TP* | *FP* | *FN* | *r* [1] | *P* [2] | *F* [3] |
| a: The images contains only one complete tree trunk | 237/474 | 114/228 | 114 | 114 | 0 | 0 | 1 | 1 | 1 |
| b: The images contains two complete tree trunks | 62/60 | 22/22 | 21 | 21 | 0 | 1 | 0.95 | 1 | 0.97 |
| c: Multiple trunks with branches appear in a voxel | 69/94 | 39/48 | 39 | 39 | 0 | 0 | 1 | 1 | 1 |
| d: The information of the trunk is occluded by leaves or branches | 36/56 | 14/18 | 14 | 14 | 0 | 0 | 1 | 1 | 1 |
| e: The information of the trunks belonging to multiple trees overlap in one voxel | 31/31 | 14/14 | 12 | 12 | 0 | 2 | 0.86 | 1 | 0.92 |

| | | | | | | | | | |
|---|---|---|---|---|---|---|---|---|---|
| f: The trunk of a tree appears in two adjacent voxels. | 54/87 | 20/29 | 19 | 19 | 3 | 1 | 0.95 | 0.86 | 0.90 |
| Overall | 489/802 | 223/359 | 219 | 219 | 3 | 4 | 0.98 | 0.99 | 0.98 |

[1] *r* (recall): tree detection rate. [2] *P* (precision): the correctness of the detected tree. [3] *F* (*F*-score): the overall accuracy of detected tree.

## 4. Discussion

### 4.1. The Advantages of Our Approach

Individual tree segmentation is still an essential part of tree property retrieval from remote sensing forest data. Previous studies [44–47] relied on single tree detection from monochromatic wavelength ALS and focused on the use of the geometric spatial information of the point clouds. However, these methods [6,46] struggled to extract clumped tree crowns of similar heights and density distributions because clumped trees do not meet the assumption of geometric constraint characteristics. For example, clumped tree crowns with similar heights and density distributions may be mistakenly detected as a single treetop [46]. Additionally, the non-treetop local maxima stemming from wild-grown branches may be falsely detected as treetops [46]. For the segmentation of individual trees based on TLS and MLS data, bottom-up or alongside scanning patterns present the phenotypic information that is mainly distributed in the lower part or along the side of a tree body because the laser sensors are mounted on a tripod or car roof that is lower than the tree crown height. Hence, primitive elements (i.e., trunks or tree crowns of lower heights) are usually taken as distinct marks to segment individual trees based on MLS and TLS data. However, the lower parts of a tree crown with many pendulous branches caused by self-weighting will disrupt the uniform shape. Trunk knots or twigs originating from a trunk will result in horizontal cross sections of trunks without well-defined circle shapes, which will complicate the location of trunks based on circle-detection algorithms such as Hough transform [48] or cylinder fitting [49]. Meanwhile, the accuracy of trunk detection algorithms based on MLS and TLS data would markedly decrease when understory vegetation is present in the forest plot, which results in the generation of much more occlusion.

Under the background of universal interferences, existing in the domains of image processing and machine vision, deep learning emerged and provided machines a greater ability to identify targets through the efficient extraction of features from vast samples and repeatedly improving the neural network performance. In reality, deep learning has been widely used in different fields to study artificial intelligence and to build intelligent systems to assist humans in various applications, such as speech recognition, image retrieval and computer predictions. Here, a projection strategy for human-portable LiDAR data combined with Faster R-CNN of deep learning was proposed to improve the segmentation of individual trees. Due to the frequent occurrence of typhoons, the rubber trees are seriously tilted, and the morphological structure of the canopy is not obvious. Therefore, individual rubber tree segmentation based on the characteristics of the canopy is difficult to achieve, and the precision of the segmentation results is susceptible to the inclination angle of rubber tree bodies and the degree of mutually occluded tree canopies.

Different from the traditional methods based on computer graphics or image processing techniques [49,50], Faster R-CNN utilises a large number of data samples to extract the semantic features of the detection target and automatically recognise the tree trunk in the deep images. As the number of training samples of tree trunks increases and the capacity of the deep learning network continues to improve, a deep learning-based algorithm with robustness, generality and scalability will appear for detecting tree trunks of different tree species and under different plot site conditions according to the framework of our algorithm concept.

### 4.2. Potential Improvement

The high accuracy obtained by our method for rubber tree trunk recognition from deep images is mainly due to the strong capacity of soil nutrient and water absorption by rubber trees. This biological property results in the absence of shrubs in the lower parts of the forest plots, which allows for the lower wood components of each rubber tree to be fully captured with minor occlusion effects. For the ecology forest plots with mixed tree plantations, due to the intricate growth of various tree species, the vigorous growth of shrubs in the lower parts of the forest plots obstructs laser scanning views and produces deficient scanned points on the target tree trunk, resulting in an increase in the complexity and noise in the generated deep images, which complicates recognition of the trunk locations. To address this issue, our follow-up work will focus on generating more training samples to strengthen the machine learning [51] and identification capacities for accurate recognition of trunks from a complex background.

The scanned point data of 3D rubber trees has multiple popular representations, leading to various approaches for learning. Pioneering studies [52–54] apply 3D convolutional neural networks on voxelised shapes and recognise the shapes of 3D objects. However, volumetric representation is constrained by its resolution due to data sparsity and the computation cost of 3D convolution. It is challenging for 3D convolutional neural networks to process very large point clouds and this approach is constrained by the representation power of the feature extraction of 3D target objects. Hence, the strategy of multi-view CNNS [55] was adopted here to render the 3D point clouds of trees into 2D images and then apply 2D convolution nets to classify them. The 2D CNN method is relatively mature and has achieved state-of-the-art performance in some image-based phenotyping tasks, with successful application in tasks such as face recognition [56], crop attribute identification [31] and video-based target tracking [57]. Practice shows that 2D CNN has more powerful performance and error-checking capacities in the current stage of deep learning development. With advances in computer hardware and target code optimisation, in the future, we will explore individual tree segmentation for different types of forest plots using collected 3D scanned points based on 3D CNN.

## 5. Conclusions

For crooked rubber trees caused by long-term hurricane disturbances, a deep-learning method based on the scanned point clouds collected by man-portable LiDAR was designed to detect the location of rubber tree trunks and accomplish individual rubber tree crown segmentation. Through the voxelisation of the scanned trunk points and projection transform from the scanned points, a total of 802 deep images providing the trunk information for three rubber tree plots of different clones was generated, which are used as the training samples for optimisation of the convolutional networks and related parameter selection. Other subsets of scanned data in three rubber tree plots were used to generate the testing samples of 359 deep images to verify the effectiveness of the convolutional neural network. The results show that our algorithm based on deep learning achieves high accuracy (> 90%) in tree trunk recognition from a large number of testing samples. Although the voxelisation with the fixed size produces multi-trunks or incomplete trunk representations in some deep images, which complicates computer understanding of trunk location in deep images, the overall accuracy of tree trunk detection still reaches 90% for all of the tested samples. Through a combination of the regional growing method and the extraction of the tree skeleton from detected tree trunks, individual tree crown segmentation for three rubber tree plots has been completed. Our work provides new guidance for forest management using artificial intelligence concepts to achieve sound performance on individual tree crown segmentation.

## 6. Patents

**Author Contributions:** Conceptualization, J.W. and X.C.; Data curation, J.W. and X.C.; Investigation, F.A., B.C. and T.Y.; Methodology, J.W., X.C. and T.Y.; Project administration, T.Y.; Resources, L.C., F.A., B.C. and L.X.; Validation, J.W. and X.C.; Writing—original draft, J.W. and X.C.; Writing—review & editing, L.C. and T.Y.

**Funding**: This work was partly supported by the National Key R&D Program of China via grant 2017YFD0600904, the National Natural Science Foundation of China via grant 31770591 and 41701510, the

National Key Research and Development Program of China via grant 2017YFD0600905-1, and the Priority Academic Program Development of Jiangsu Higher Education Institutions (PAPD) via grant 1111.

**Conflicts of Interest:** The authors declare no conflict of interest.

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
