# Peer review of "Individual Rubber Tree Segmentation Based on Ground-Based LiDAR Data and Faster R-CNN of Deep Learning"

_forests, doi:10.3390/f10090793_

Round 1

Reviewer 1 Report

The authors present a novel contribution for individual tree segmentation based on transformation of voxels into 2D images for deep learning based trunk detection.

The methods show great promise for automated tree detection particularly in plantations.

Challengues for heterogenoeus stands with compelx undertory are acknowledged, as well as the need for further sampling in future studies to continue improving the algorithm training.

In general, methods are clearly presented, although the article might benefit from a clearer reorganization, moving methodological paragraphs from the results section to the methods section.

Introduction and discussion should also better present existing previous studies with TLS and how those previous approaches differ from the current study.

Specific comments.

Introduction.

Line 44. Please add a reference.

The authors first introduce the types of LIDAR sensors, (lines 53-66) and techinques, without clearly diferenciating which studies are TLS or ALS (refs 7-13) or focusing on ALS only (refs 14-19)
I would recommend a clearer presentation of ALS and TLS previous studies in the introduction.
More focus should be put into discussing previous TLS studies, since it is the focus of the current study.
For example, line 85 mentions "a few pioneering studies" of tree trunk detection with TLS, but does not contain the references to those TLS studies.
A summary of the findings, limitations, and needs for future work based on previous TLS studies is needed.
Also, I would recommend to highlight in the introduction what is the difference of the current approach with those previous work to stress the originality of the proposed approach.

Methods.
Please define "anchor" so that the reader doesn't have to search for reference 32 to interpret this parameter.

Results.
Several paragraphs (e.g. 323-330, 346-351, 380-386) probably would better be presented before in the methodology section, so that this section presents results only.

Discussion.
In section 4.1., only ALS studies are mentioned. The authors should include a discussion of the findings of the current study in the context of previous TLS studies.
For example, what are the typical accuracies found in automated trunk detection?
What are the computer requirements mentioned in previous studies to run 3D classification, and how does it compare with the 2D methodology presented here?
What are the main innovations of this methodology compared to those previous studies?

Author Response

The detailed responses to the reviewer's comments are recorded in the attached file.

Reviewer 2 Report

This paper presents a tree segmentation technique which accurately identifies tree trunks using an adapted convolutional neural network. The trunks are then used as seed points for segmenting individual tree crowns in the LiDAR point cloud.

The paper is well written and the results for study region are convincing.

The authors demonstrate that this new method can be successfully applied for detecting rubber trees in plantation forests without undergrowth. It is unfortunate that it is not within the scope of the paper to assess the performance of the algorithm in more complex forests with undergrowth.

Comments:

The main objective of this study is to segment individual trees (including the tree crown) in the LiDAR point cloud. However, the manuscript mainly focuses on accurately identifying tree trunks. Please add a section with more information on the region-growing approach of classifying the final point clouds as depicted in Figure 10.

Abstract: It is not clear what deep images mean in this context (it is basically a normal image used for deep learning?) and please state the size of the voxel.  Please also clarify which deep learning method is used (adapted CNNs).

Line 90: Please add reference and give a bit of context on deep learning (what is different to other ML techniques, hidden layers etc.)

Line 94: CNNs are not necessarily the ‘typical method’ for any deep learning algorithm. You mean image recognition here specifically?

Line 102: Please clarify what a ‘deep image’ is and what it makes them massive. From what I understand they are reasonably small images used for deep learning / CNN?

Line: 103: Please replace “deep learning method” throughout the paper with a more concrete phrasing, or define the method at the beginning

Figure 1: Spacial Projection -> Spatial Projection

Line 104: Workflow Figure 1 better fits to Method section, not Introduction

Line 145: Why is the speed information useful? This is probably just a very rough average?

Line 147: Is GPS used for real-time geo-referencing of the laser returns / generated point cloud during acquisition?

Line 174: Please add reference and explanation

Line 284: Please add references to loss function of the training process and/or detail what your adaptions were

Line 318: How do you explain the strong fluctuations in the training loss (Figure 6)?

Author Response

(The authors gave the same response as above.)
